# Modelling of Irreversible Homogeneous Reaction on Finite Diffusion Layers

**Singaravel Anandhar Salai Sivasundari** [1], **Rathinam Senthamarai** [2], **Mohan Chitra Devi** [3], **Lakshmanan Rajendran** [1,*] **and Michael E. G. Lyons** [4,*]

1. Department of Mathematics, AMET University, Chennai 603112, Tamilnadu, India
2. Department of Mathematics, College of Engineering and Technology, SRM Institute of Science and Technology, Chennai 603203, Tamilnadu, India
3. Department of Mathematics, University College of Enginnering, Anna University, Dindigul 624622, Tamilnadu, India
4. School of Chemistry & AMBER National Centre, Trinity College Dublin, University of Dublin, D02 PN40 Dublin, Ireland
* Correspondence: raj_sms@rediffmail.com (L.R.); melyons@tcd.ie (M.E.G.L.)

**Abstract:** The mathematical model proposed by Chapman and Antano (Electrochimica Acta, 56 (2010), 128–132) for the catalytic electrochemical–chemical (EC') processes in an irreversible second-order homogeneous reaction in a microelectrode is discussed. The mass-transfer boundary layer neighbouring an electrode can contribute to the electrode's measured AC impedance. This model can be used to analyse membrane-transport studies and other instances of ionic transport in semiconductors and other materials. Two efficient and easily accessible analytical techniques, AGM and DTM, were used to solve the steady-state non-linear diffusion equation's infinite layers. Herein, we present the generalized approximate analytical solution for the solute, product, and reactant concentrations and current for the small experimental values of kinetic and diffusion parameters. Using the Matlab/Scilab program, we also derive the numerical solution to this problem. The comparison of the analytical and numerical/computational results reveals a satisfactory level of agreement.

**Keywords:** mathematical modeling; irreversible homogeneous reaction; Akbari-Ganji method; differential transform method; reaction-diffusion equations

## 1. Introduction

The role of electrochemical impedance spectroscopy in electrode processes is crucial. It is a useful experimental technique employed in an electrode procedure to categorise different mechanisms [1]. In the semi-infinite system, the diffusion-controlled resistance also contributes to the measured impedance. The phase at the electrode surface can have its diffusion impedance determined using the transport model [2–6]. Several authors [7–11] studied reactant diffusion through a stagnant diffusion layer of thickness. Juan Bisquert [12] discussed the theory of electron diffusion and recombination impedance in a thin layer. Ten years ago, Chapman and Antano [1] used a computational approach to find the approximate concentration profiles and impedance behaviour. Uma et al. [13] derived the approximate analytical expressions for the concentration of the system.

To the best of our knowledge, there is no concise and closed-form analytical equation provided for species concentrations in irreversible homogeneous reactions in finite-layer diffusion. This study intends to obtain new analytical expressions, in closed form, for the concentration of the reactant *S*, product *P*, and solute *R* for low values of the rate constant.

## 2. Mathematical Formulation

The following describes the reaction mechanism for a catalytic electrochemical system with diffusion and an irreversible second-order reaction in a stagnant diffusion layer [1]:

$$R \pm e^- \leftrightarrow P \tag{1}$$

$$P + S \xrightarrow{k_2} Y + R \tag{2}$$

The soluble substance $P$ is produced at the electrode by the oxidation or reduction of the solute $R$. With the reaction rate constant, $P$ reacts irreversibly in the solution to create the product $Y$ and regenerate $R$ from the electrochemically inactive reactant $S$ that is already present in the bulk solution. Figure 1 depicts the overview of a second-order irreversible homogeneous reaction. The overall process involves the electrochemical conversion of $S$ to $Y$, which is accelerated by $R$, with some accumulation of $P$, if the homogeneous reaction cannot completely utilise the material produced at the electrode. One particular example is the oxidation of sulfite ($S$) to sulphate ($Y$), which is accelerated by the ferrous ion ($R$) and results in the formation of a reactive ferric ion ($P$) at an anode.

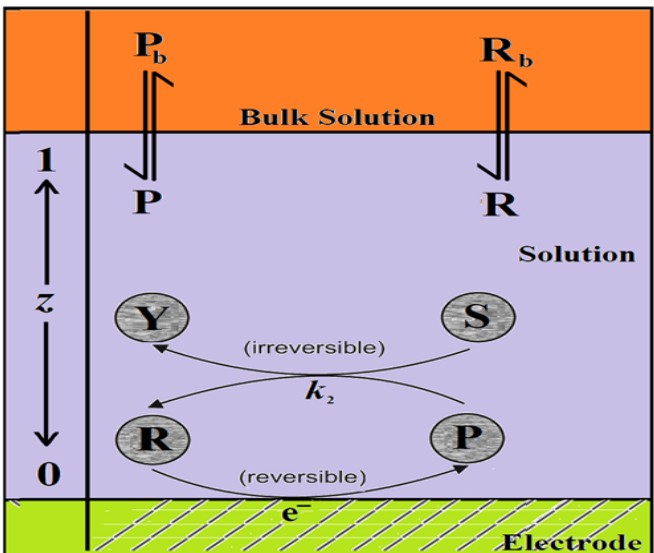

**Figure 1.** General scheme of a second-order irreversible homogeneous reaction.

Consider the non-linear differential equations [1] at steady-state conditions in a dimensionless form, as follows:

$$\frac{d^2 R(x)}{dx^2} + k\, P(x)\, S(x) = 0 \tag{3}$$

$$\frac{d^2 P(x)}{dx^2} - k\, P(x)\, S(x) = 0 \tag{4}$$

$$\frac{d^2 S(x)}{dx^2} - k\, P(x)\, S(x) = 0 \tag{5}$$

where the dimensionless variables are

$$R = [C_R/C_{Sb}], P = [C_P/C_{Sb}], S = [C_S/C_{Sb}], x = z/\delta$$
$$k = [k_2 C_{Sb}/\delta^2 D], \alpha = [C_{Rb}/C_{Sb}] \text{ and } \gamma = [C_{R0,SS}/C_{Sb}] \tag{6}$$

where $R$, $P$, and $S$ are the dimensionless concentrations solute, product, and the reactant, respectively; $x$ is the dimensionless distance. For simplicity, we have assumed that all three diffusion coefficients are equal to the value $D$; $k$ is dimensionless rate constant. Two other

parameters, $\alpha$ and $\gamma$, are the concentration ratios. The corresponding boundary conditions are as follows:

$$R = \gamma; P = \alpha - \gamma; \frac{dS}{dx} = 0 \text{ when } x = 0 \tag{7}$$

$$R = \alpha; P = 0; S = 1 \text{ when } x = 1 \tag{8}$$

where $\alpha > \gamma$. The non-dimensional current is given as

$$\psi = \frac{i\delta}{nFDC_{Sb}A} = \left| \frac{dR(x)}{dx} \right|_{x=0} \tag{9}$$

## 3. Analytical Expression of Concentrations

Equations (3)–(5) represent the nonlinear differential equations. Finding the precise solution to these nonlinear differential equations is difficult. One of the toughest challenges, especially across a wide range of science and engineering applications, is solving nonlinear differential equations. Recently, the construction of an analytical solution has been the focus of numerous analytical techniques, such as the homotopy perturbation method (HPM) [14–26], the variational iteration method [27–32], the homotopy analysis method [33–37], the Akbari-Ganji method [38–43], the Taylor series method [44–47], and the differential transform method. Jalili et al. [48–50] discussed the heat exchange in nanoparticles and solved the momentum and energy equation numerically. In this paper, AGM and DTM are developed (Appendices A and B) for solving the ill-posed boundary value problem, which has the fewest number of unknowns, and its associated boundary conditions are represented by the Equations (3)–(8). For the nonlinear steady-state second-order equations, these are effective techniques.

### 3.1. Analytical Expression of Concentrations Using the Akbari-Ganji Method (AGM)

The Akbari-Ganji approach, created by the mathematicians Akbari and Ganji [38–43], is used in this study to solve the nonlinear differential equations governing this system. Moreover, with this method, we can quickly solve the nonlinear equations without any complex mathematical operations. The relation between the concentrations is given in Appendix A. Using this relationship and the proposed approach, the general analytical expression for normalized concentrations of the species are as follows (Appendix B):

$$R(x) = -\frac{cosh(mx)}{cosh(m)} + (\alpha + 1)\, x - (\gamma + l)(x - 1) \tag{10}$$

$$P(x) = \frac{cosh(mx)}{cosh(m)} - (\alpha + 1)\, x + (\gamma + l)(x - 1) + \alpha \tag{11}$$

$$S(x) = \frac{cosh\, mx}{cosh\, m} \tag{12}$$

where the constant

$$m = \sqrt{k(\alpha - \gamma)} \tag{13}$$

Using Equation (7), the non-dimensional current is

$$= \frac{i\delta}{nFDC_{Sb}A} = \left| \frac{dR(x)}{dx} \right|_{x=0} = 1 + \alpha - \gamma - sech(m) \tag{14}$$

### 3.2. Analytical Expression of Concentrations Using the Differential Transform Method (DTM)

A semi-analytical technique for resolving differential equations is the differential transform method (DTM). Zhou [35] was the first to put forth the differential transform idea, which is used to address both linear and nonlinear boundary value issues in electric circuit analysis. The nth derivative of an analytical function at a particular point can be precisely calculated using DTM, regardless of whether the boundary conditions are known

or unknown. With this method, differential equations produce an empirical polynomial solution. This approach differs from the typical high-order Taylor series process, which requires the symbolic computation of the data functions. The Taylor series procedure takes some time to compute. The DTM is an alternative iterative procedure for obtaining analytical solutions of differential equations [36–39]. The approximate analytical expressions of concentrations using the DTM method are obtained as follows (Appendix C):

$$R(x) = (\alpha + 1)\, x - \left( \gamma + \frac{2}{2 + k(\alpha - \gamma)} \right) (x - 1) - \frac{2 + k\,(\alpha - \gamma)x^2}{2 + k(\alpha - \gamma)} \tag{15}$$

$$P(x) = \alpha - (\alpha + 1)\, x + \left( \gamma + \frac{2}{2 + k(\alpha - \gamma)} \right) (x - 1) + \frac{2 + k\,(\alpha - \gamma)x^2}{2 + k(\alpha - \gamma)} \tag{16}$$

$$S(x) = \frac{2 + k\,(\alpha - \gamma)x^2}{2 + k(\alpha - \gamma)} \tag{17}$$

## 4. Validation of Analytical Results with Numerical Simulation

The validation method has received significant attention in the literature. Using the function pdex4 in the Scilab software, the nonlinear differential Equations (3)–(5), with the boundary conditions (7) and (8), are numerically solved. The Scilab code is also given in Appendix D. Figures 2 and 3 compare the species concentrations obtained using the AGM technique, Equations (10)–(12), and the DTM method, Equations (15)–(17), with a numerical solution. Figure 4 represents the dimensionless current versus the dimensionless rate constant $k$. Tables 1 and 2 show the comparison between the numerical and analytical concentration of the substrate obtained by AGM and DTM for various values of parameters, $\alpha = 0.8$, $\gamma = 0.5$, and for different values of $k$. From the tables it is found that the average relative errors are less than 5%. When considering the concentrations of the solute $R$, product $P$, and reactant $S$ for small values of other parameters, our analytical results show satisfactory agreement for $k \leq 1$.

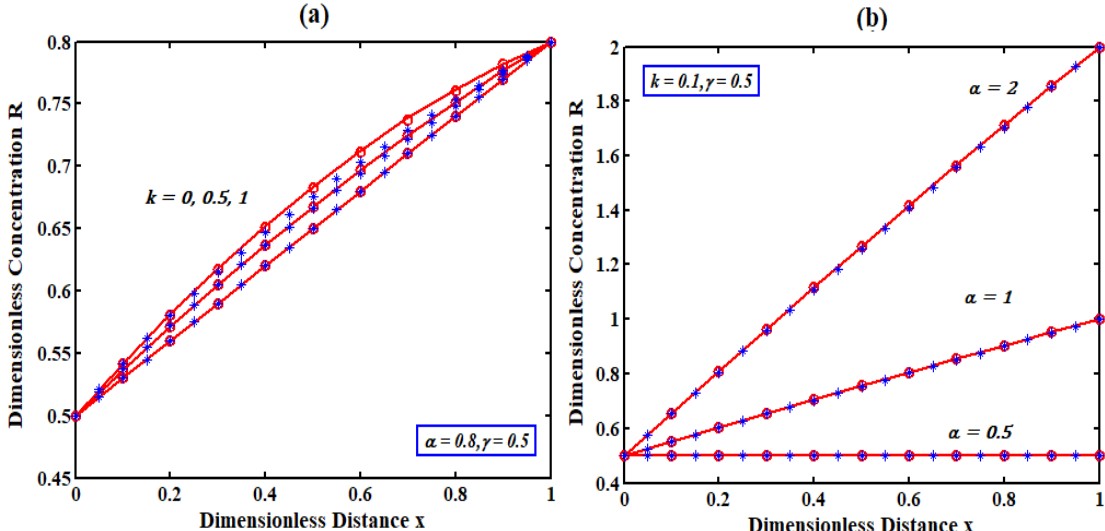

**Figure 2.** *Cont.*

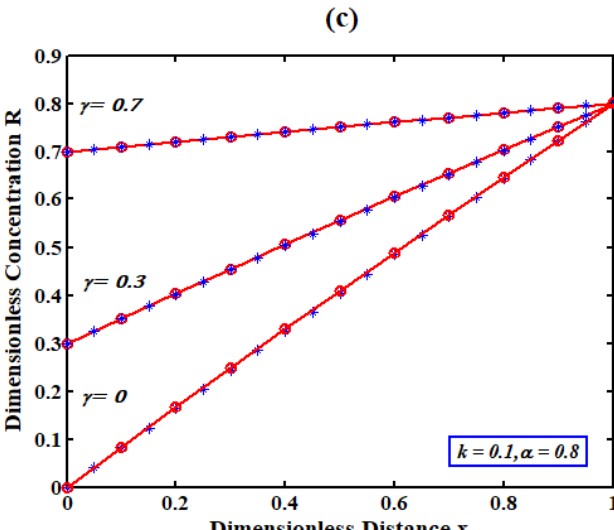

**Figure 2.** (**a**–**c**) Profile of the normalized steady-state concentrations *R* versus the normalized distance *x* for various values of the parameters *k*, *α*, and *γ* using Equations (10) and (15) The solid line denotes the AGM method, (o o o) represents DTM, and (* * *) denotes numerical simulation.

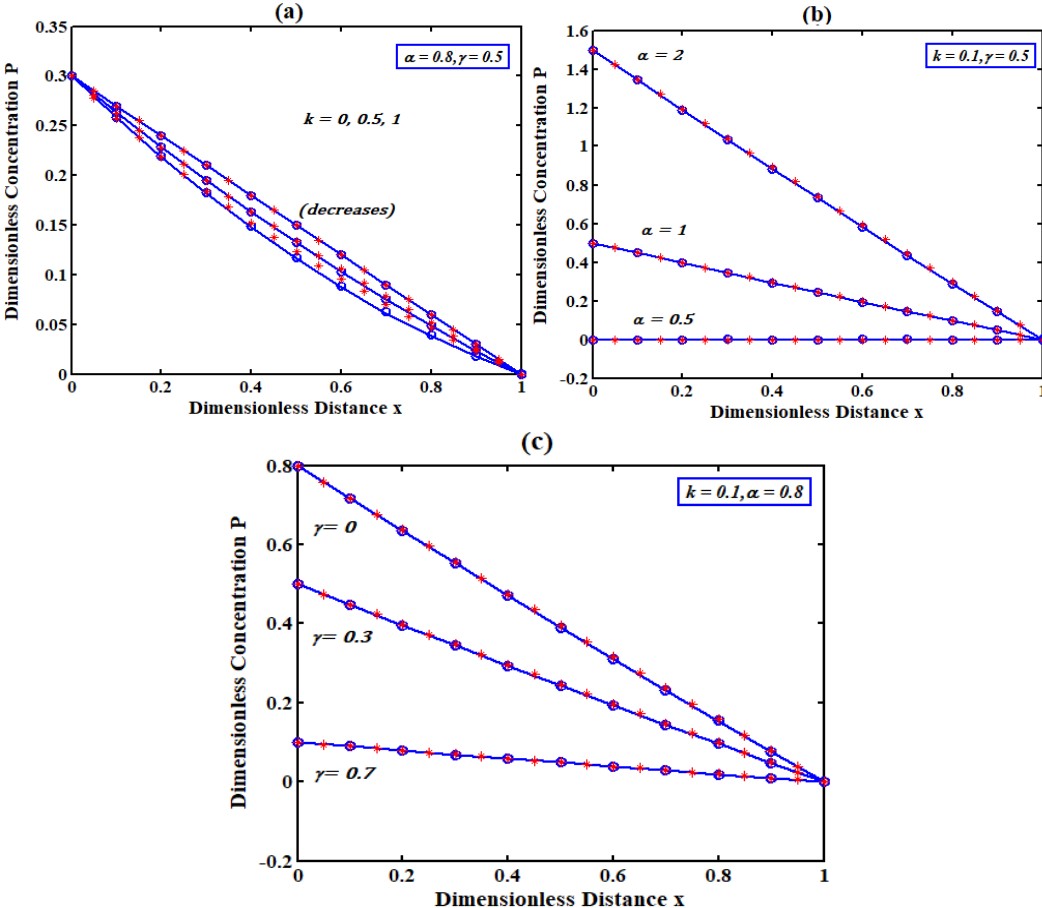

**Figure 3.** (**a**–**c**): Profile of the normalized steady-state concentrations *P* versus the normalized distance *x* for various values of the parameters *k*, *α*, and *γ* using Equations (11) and (16). The solid line denotes the AGM method, (o o o) represents DTM, and (* * *) denotes numerical simulation.

**Table 1.** Comparison between the numerical and analytical expression of the concentration of the substrate obtained by AGM and DTM for the parameters $\alpha = 0.8, \gamma = 0.5,$ and for different values of $k$.

| | Substrate Concentration S | | | | | | | | | | | | | | |
| | $k=0$ | | | | | $k=0.5$ | | | | | $k=1$ | | | | |
| $x$ | NUM | AGM Equation (12) | DTM Equation (17) | Error % for AGM | Error % for DTM | NUM | AGM Equation (12) | DTM Equation (17) | Error % for AGM | Error % for DTM | NUM | AGM Equation (12) | DTM Equation (17) | Error % for AGM | Error % for DTM |
|---|---|---|---|---|---|---|---|---|---|---|---|---|---|---|---|
| 0 | 1 | 1 | 1 | 0 | 0 | 0.9535 | 0.9294 | 0.9302 | 2.53 | 2.44 | 0.913 | 0.8667 | 0.8696 | 5.07 | 4.75 |
| 0.25 | 1 | 1 | 1 | 0 | 0 | 0.9576 | 0.9338 | 0.9346 | 2.49 | 2.40 | 0.9206 | 0.8748 | 0.8777 | 4.98 | 4.66 |
| 0.5 | 1 | 1 | 1 | 0 | 0 | 0.9681 | 0.9469 | 0.9477 | 2.19 | 2.11 | 0.9405 | 0.8994 | 0.9022 | 4.37 | 4.07 |
| 0.75 | 1 | 1 | 1 | 0 | 0 | 0.9830 | 0.9689 | 0.9695 | 1.43 | 1.37 | 0.9683 | 0.9409 | 0.9429 | 2.83 | 2.62 |
| 1 | 1 | 1 | 1 | 0 | 0 | 1.0000 | 1.0000 | 1.0000 | 0.00 | 0.00 | 1.0000 | 1.0000 | 1.0000 | 0.00 | 0.00 |
| | Average Error % | | | 0 | 0 | Average Error % | | | 1.7274 | 1.6652 | Average Error % | | | 3.4481 | 3.2220 |

**Table 2.** Comparison between the numerical and analytical expression of the concentration of the substrate obtained by AGM and DTM for the parameters $k = 0.1, \gamma = 0.5,$ and for different values of $\alpha$.

| | Substrate Concentration S | | | | | | | | | | | | | | |
| | $\alpha=0.5$ | | | | | $\alpha=1$ | | | | | $\alpha=2$ | | | | |
| $x$ | NUM | AGM Equation (12) | DTM Equation (17) | Error % for AGM | Error % for DTM | NUM | AGM Equation (12) | DTM Equation (17) | Error % for AGM | Error % for DTM | NUM | AGM Equation (12) | DTM Equation (17) | Error % for AGM | Error % for DTM |
|---|---|---|---|---|---|---|---|---|---|---|---|---|---|---|---|
| 0 | 1 | 1 | 1 | 0 | 0 | 0.9837 | 0.9755 | 0.9756 | 0.83 | 0.82 | 0.9524 | 0.9294 | 0.9302 | 2.41 | 2.33 |
| 0.25 | 1 | 1 | 1 | 0 | 0 | 0.9851 | 0.9770 | 0.9771 | 0.82 | 0.81 | 0.9565 | 0.9338 | 0.9346 | 2.37 | 2.29 |
| 0.5 | 1 | 1 | 1 | 0 | 0 | 0.9888 | 0.9816 | 0.9817 | 0.73 | 0.72 | 0.9673 | 0.9469 | 0.9477 | 2.11 | 2.03 |
| 0.75 | 1 | 1 | 1 | 0 | 0 | 0.9940 | 0.9893 | 0.9893 | 0.47 | 0.47 | 0.9825 | 0.9689 | 0.9695 | 1.38 | 1.32 |
| 1 | 1 | 1 | 1 | 0 | 0 | 1.0000 | 1.0000 | 1.0000 | 0.00 | 0.00 | 1.0000 | 1.0000 | 1.0000 | 0.00 | 0.00 |
| | Average Error % | | | 0 | 0 | Average Error % | | | 0.5714 | 0.5653 | Average Error % | | | 1.6563 | 1.5940 |

NUM—numerical simulation; AGM—Akbari-Ganji method; DTM—differential transform method.

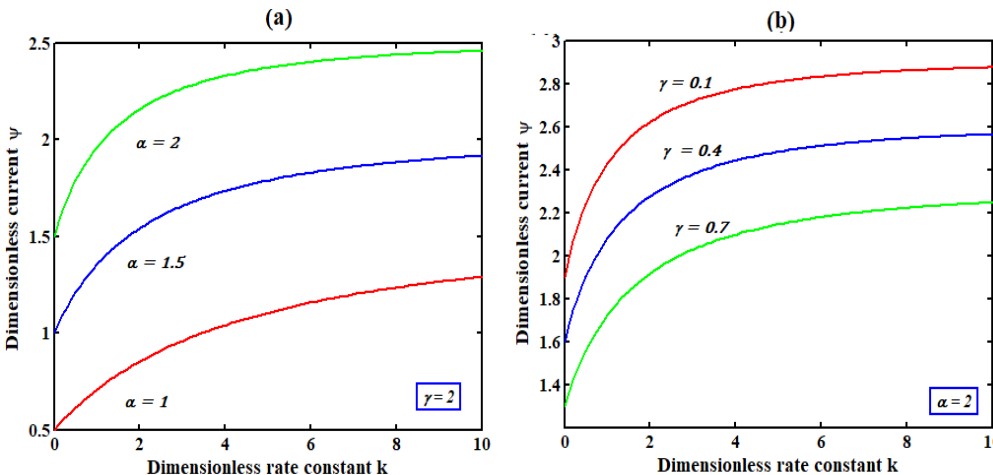

**Figure 4.** (**a**,**b**): Dimensionless current versus dimensionless rate constant *k*.

## 5. Discussions

Equations (10)–(12) and (15)–(17) are the new simple analytical expressions of the concentrations of the solute *R*, product *P*, and the reactant *S*, respectively. The concentration profiles depend upon diffusion parameters (*α* and *γ*) and rate constant (*k*). In Figure 2a,c, the profile of the solute concentration is presented. These figures clearly show that for all small feasible values of the parameters, the solute concentration *R* increases at the electrode surfaces, whereas it drops as *γ* and *k* increase. Product concentration *P* increases in Figure 3a,c when the parameters *k* and alpha increase, while it increases when *γ* decreases. However, there is no discernible difference between the variances of the parameters *k*, *α*, and *γ* and the reactant concentration *S*. It has been noted that a rise in the rate constant *k* causes a fall in the *S* concentration. Figure 4a,b plots the current against parameter *k*. The figure indicates that the current increases as the rate constant *k* rises.

## 6. Conclusions

The set of nonlinear equations in the irreversible homogeneous reaction for finite diffusion is discussed in this study. Common analytical expressions are provided for the concentration of solute, product, reactant, and current for all diffusion values and the small values of the kinetic parameters. Compared with other analytical methods, the AGM and DTM are straightforward, with a simple solution processes, yielding accurate results. The non-steady-state circumstances can also be handled with these methods. The theoretical and numerical results were further contrasted, and they were found to be in good accord. The method described here can be applied to examine membrane-transport studies, as well as some other instances of ionic transport in semi-conductors and solid electrolytes. This theoretical approach could be applied in more complicated cases when the transport equation contains non-linearities. It can also be used with membranes and solid electrolytes, where diffusion is a crucial phenomenon.

**Author Contributions:** S.A.S.S.: data creation, software, formal analysis, writing—original draft; R.S.: writing, formal analysis, investigation, resources; M.C.D.: formal analysis, validation, visualization; L.R.: conceptualization, methodology, project administration, supervision, writing—review and editing; M.E.G.L.: investigation, resources, supervision, writing—review and editing. All authors have read and agreed to the published version of the manuscript.

**Funding:** This research received no external funding.

**Institutional Review Board Statement:** Not applicable.

**Informed Consent Statement:** Not applicable.

**Data Availability Statement:** Not applicable.

**Acknowledgments:** This study was supported by AMET, a deemed university, Chennai. The authors are grateful for the support of Shri J. Ramachandran, G. Thiruvasagam, and M. Jayaprakashvel, Registrar, AMET, a deemed university, Chennai, Tamil Nadu, India. The authors are also very grateful to the management of the SRM Institute of Science and Technology, Kattankulathur, India, for their continuous support and encouragement.

**Conflicts of Interest:** The authors declare no conflict of interest.

## Nomenclature

| Symbols | Name | Unit |
|---|---|---|
| $C_R$ | Concentration of reactant | Mol cm$^{-3}$ |
| $C_P$ | Concentration of product | Mol cm$^{-3}$ |
| $C_S$ | Concentration of solute | Mol cm$^{-3}$ |
| $C_{Rb}$, $C_{Sb}$ | Bulk concentration | Mol cm$^{-3}$ |
| $C_{R0,SS}$ | Concentration of $R$ at the electrode in steady-state | Mol cm$^{-3}$ |
| $\delta$ | Diffusion layer thickness | cm |
| $D$ | Diffusion coefficient | cm$^2$s$^{-1}$ |
| $k_2$ | Reaction-rate constant | Mol cm$^{-3}$s |
| $z$ | Distance from the electrode surface | cm |
| $R$ | Dimensionless concentration of reactant | None |
| $P$ | Dimensionless concentration of product | None |
| $S$ | Dimensionless concentration of solute | None |
| $x$ | Dimensionless distance | None |
| $k$ | Dimensionless rate constant | None |
| $\alpha$, $\gamma$ | Concentration ratio | None |
| $\psi$ | Dimensionless current | None |
| $n$ | Number of electrons transferred | None |

## Appendix A. The Relationship between Concentrations of Species

In irreversible homogeneous reactions, the nonlinear second-order differential Equations (3)–(5) are as follows:

$$\frac{d^2R(x)}{dx^2} + k\,P(x)\,S(x) = 0 \tag{A1}$$

$$\frac{d^2P(x)}{dx^2} - k\,P(x)\,S(x) = 0 \tag{A2}$$

$$\frac{d^2S(x)}{dx^2} - k\,P(x)\,S(x) = 0 \tag{A3}$$

The boundary conditions are

$$R = \gamma; P = \alpha - \gamma; \frac{dS}{dx} = 0 \text{ when} x = 0 \tag{A4}$$

$$R = \alpha; P = 0; S = 1 \text{ when} x = 1 \tag{A5}$$

Adding Equations (A1)–(A3), we get,

$$\frac{d^2R}{dx^2} + \frac{d^2S}{dx^2} = 0 \tag{A6}$$

Integrating (A6) twice, we get,

$$R(x) = -S(x) + C_1 x + C_2 \tag{A7}$$

Using the boundary conditions (A4) and (A5) and simplifying, we obtain the relation as follows:

$$R(x) = -S(x) + (\alpha + 1)\,x - (\gamma + S(0))(x - 1) \tag{A8}$$

where $S(0) = S(x = 0)$ is obtained using AGM and DTM.

Subtracting Equations (A2) and (A3), we get,

$$\frac{d^2P}{dx^2} - \frac{d^2S}{dx^2} = 0 \tag{A9}$$

Integrating (A9) twice, we get,

$$P(x) = S(x) + C_1 x + C_2 \tag{A10}$$

Using the boundary conditions (A4) and (A5) and simplifying, we obtain the relationship as follows:

$$P(x) = S(x) - (\alpha + 1)\,x + (\gamma + S(0))(x - 1) + \alpha \tag{A11}$$

**Appendix B. Analytical Solution of the Equations (3)–(5) Using AGM**

The system of non-linear second-order differential Equations (3)–(5) in irreversible homogeneous reactions are given as follows:

$$\frac{d^2R(x)}{dx^2} + k\,P(x)\,S(x) = 0 \tag{A12}$$

$$\frac{d^2P(x)}{dx^2} - k\,P(x)\,S(x) = 0 \tag{A13}$$

$$\frac{d^2S(x)}{dx^2} - k\,P(x)\,S(x) = 0 \tag{A14}$$

$$R = \gamma; P = \alpha - \gamma; \frac{dS}{dx} = 0 \text{ when } x = 0 \tag{A15}$$

The boundary conditions are

$$R = \alpha; P = 0; S = 1 \text{ when } x = 1 \tag{A16}$$

Using the relation between $P$ and $S$ (A10), the Equation (A14) can be written as follows

$$\frac{d^2S}{dx^2} - k(S - (\alpha + 1)\,x + (\gamma + S(0))(x - 1) + \alpha)S = 0 \tag{A17}$$

By using the AGM method, we consider the trial solution

$$S(x) = A\cosh mx + B\sinh mx \tag{A18}$$

where $A,\ B$ and $m$ are constants.

Using the boundary conditions (A15) and (A16) in (A18),we get

$$B = 0,\ A = \frac{1}{\cosh m} \tag{A19}$$

By substituting (A19) in (A18), we get

$$S(x) = \frac{cosh\ mx}{cosh\ m} \tag{A20}$$

By using AGM, the value of '$m$' can be obtained as follows:
Substitute (A20) in (A17), and we get

$$m^2\frac{cosh\ mx}{cosh\ m} - k\left(\frac{cosh\ mx}{cosh\ m} - (\alpha+1)\,x + (\gamma + sech(m))(x-1) + \alpha\right)\frac{cosh\ mx}{cosh\ m} = 0 \tag{A21}$$

By substituting $x = 0$ in (A21) and simplifying, we get

$$\frac{m^2 - k\alpha + k\gamma}{cosh(m)} = 0 \tag{A22}$$

$m$ can be obtained from the above equation as follows

$$m = \sqrt{k(\alpha - \gamma)} \tag{A23}$$

where $\alpha - \gamma > 0$.

Using (A8) and (A11), we get the analytical expressions of concentrations of $R$ and $P$, which are given in the main text Equations (10)–(12).

## Appendix C. Approximate Analytical Solution of Nonlinear Differential Equations (3)–(5) Using the DTM

Consider the differential equation and boundary conditions

$$\frac{d^2S(x)}{dx^2} - k\,P(x)\,S(x) = 0 \tag{A24}$$

$$P = \alpha - \gamma;\ \frac{dS}{dx} = 0\ \text{when}\ x = 0 \tag{A25}$$

The transformed version of (A24) and (A25) are, respectively, given by

$$(n+2)(n+1)\,S(n+2) - k\sum_{r=0}^{n} S(n)\,P(n-r) = 0 \tag{A26}$$

$$P(0) = \alpha - \gamma, S(1) = 0. \tag{A27}$$

Assume that

$$S(0) = l \tag{A28}$$

Letting $n = 0$ and substituting (A27) and (A28) into (A26), imply

$$2\,S(2) - k(S(0)\,P(0)) = 0, \tag{A29}$$

that is,

$$S(2) = \frac{k\,l\,(\alpha - \gamma)}{2} \tag{A30}$$

The differential inverse transforms of $u(n)$ is defined as

$$S(x) = \sum_{m=0}^{2} S(n)(x - x_0)^n, \tag{A31}$$

By letting $x_0 = 0$, we obtain the following second-order closed-form solution

$$S(x) = \sum_{m=0}^{2} S(n)(x)^n = l + \frac{k\,l\,(\alpha - \gamma)}{2}x^2 \tag{A32}$$

By using the boundary conditions $S = 1$ when x = 1, we get the value of $l$ as

$$l = \frac{2}{2 + k(\alpha - \gamma)},$$ (A33)

and hence, the approximate analytical solution for the concentration of the substrate is

$$S(x) = \frac{2 + k(\alpha - \gamma)x^2}{2 + k(\alpha - \gamma)}$$ (A34)

**Appendix D. Numerical Solution of Nonlinear Equations (3)–(5)**

```
function pdex4
m = 0;
x = linspace(0,1);
t = linspace(0,1000);
sol = pdepe(m,@pdex4pde,@pdex4ic,@pdex4bc,x,t);
u1 = sol(:,:,1);
u2 = sol(:,:,2);
u3 = sol(:,:,3);
figure
plot(x,u1(end,:))
title('u1(x,t)')
figure
plot(x,u2(end,:))
title('u2(x,t)')
figure
plot(x, u3(end,:))
title('u3(x,t)')
% —————————————————————————————
function [c,f,s] = pdex4pde(x,t,u,DuDx)
κ = 0.5; %These parameter values are used in Figure 2
c = [1;1;1];
f = [1;1;1].*DuDx;
F1 = κ * u(2)*u(3);
F2 = -κ *u(2)*u(3);
F3 = -κ *u(2)*u(3);
s = [F1;F2;F3];
% —————————————————————————————
function u0 = pdex4ic(x)
u0 = [0; 0; 1];
% —————————————————————————————
function [pl,ql,pr,qr] = pdex4bc(xl,ul,xr,ur,t)
pl = [ul(1)-0.5;ul(2)-0.3;0];
ql = [0;0;1];
pr = [ur(1)-0.8;ur(2);ur(3)-1];
qr = [0; 0; 0];
```

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
