# Peer review of "Modelling of Irreversible Homogeneous Reaction on Finite Diffusion Layers"

_2673-3293, doi:10.3390/electrochem3030033_

Round 1

Reviewer 1 Report

1. What is the application of this work? (It should be added to the Abstract)

2. Some typo errors in the article should be corrected.

3. It is recommended to add some suggestions for future works in this area to improve the conclusion.

4. The Abstract should answer the following questions: What problem was studied, and why is it important? What methods were used? What are the significant results?

5. The "Introduction" section should be impressive and informative. Some relative papers in the analytical field are highly recommended to improve this section.

10.1016/j.csite.2022.102086

10.1016/j.csite.2022.102209

10.1002/htj.22582

International Journal of Material Science Innovations, 2014, 2(1), 8-17

6. The mathematical equations must be double-checked and write in word format.

7. The quality of the figures is not suitable and should be shown larger.

Reviewer 2 Report

please see attached comments

Reviewer 3 Report

Analytical solutions/expressions are always welcome, but solely, without any deeper analysis are too small to contribute. In this form, the content of the article is small for a publication. I would suggest improving it by:

- adding stability analysis and comparing it with your results. Stability analysis must include all constants in the system.

- this is a boundary value problem, so in general, it is an ill-posed problem. Can you improve the model having a pure initial value problem?

- compare different analytical method solutions and try to find some new conclusions

Besides the upper additions to the article structure, it should also be improved:

 - describe using some physical explanation, what system (3)-(5) means

- equations should not be seen as images but as text (vector format)

- system (3)-(5) rewrite to (3) -> (3.1)-(3.3) and boundary conditions (7)-(8) to (7.1)-(7.2), latter your refer to system (3) with BC (7)!

- what about the rest of the variables Csr, …, not included in the nomenclature, you must also describe (I suppose that are saturated values?)

Round 2

Reviewer 1 Report

The authors have responded to my previous comments and revised the manuscript accordingly. I believe it is acceptable for publication in its form

Reviewer 3 Report

Fundamental comments are not improved. Even the structure of the article is not sufficient.